# The Continuum of Metastatic Prostate Cancer: Interpreting PSMA PET Findings in Recurrent Prostate Cancer

**DOI:** 10.3390/cancers14061361

**Published:** 2022-03-08

**Authors:** Adam M. Kase, Winston Tan, John A. Copland, Hancheng Cai, Ephraim E. Parent, Ravi A. Madan

**Affiliations:** 1Division of Hematology Oncology, Mayo Clinic, 4500 San Pablo Road, Jacksonville, FL 32224, USA; tan.winston@mayo.edu; 2Cancer Biology Department, Mayo Clinic, 4500 San Pablo Road, Jacksonville, FL 32224, USA; copland.john@mayo.edu; 3Nuclear Medicine Division of Radiology Department, Mayo Clinic, 4500 San Pablo Road, Jacksonville, FL 32224, USA; cai.hancheng@mayo.edu (H.C.); parent.ephraim@mayo.edu (E.E.P.); 4Genitourinary Malignancies Branch, Center for Cancer Research, National Cancer Institute, 10 Center Drive, 13n240b, Bethesda, MD 20892, USA; madanr@mail.nih.gov

**Keywords:** biochemical recurrent prostate cancer, PSMA PET–CT, molecular imaging, recurrent prostate cancer, metastatic prostate cancer, conventional imaging

## Abstract

**Simple Summary:**

Molecular imaging with PSMA PET–CT is more accurate and sensitive than conventional imaging with CT, MRI, and a Technetium-99 bone scan. This new imaging modality will result in more advanced disease being diagnosed earlier which may improve survival, however, it could also lead to overtreatment. Since molecular imaging has the potential to identify disease prior to its detection on conventional imaging, this highlights that advanced prostate cancer exists on a continuum. This review discusses how PSMA PET–CT can be used in managing prostate cancer using clinical scenarios.

**Abstract:**

Conventional imaging has been the standard imaging modality for assessing prostate cancer recurrence and is utilized to determine treatment response to therapy. Molecular imaging with PSMA PET–CT has proven to be more accurate, sensitive, and specific at identifying pelvic or distant metastatic disease, resulting in earlier diagnosis of advanced disease. Since advanced disease may not be seen on conventional imaging, due to its lower sensitivity, but can be identified by molecular imaging, this reveals that metastatic prostate cancer occurs on a continuum from negative PSMA PET–CT and negative conventional imaging to positive PSMA PET–CT and positive conventional imaging. Understanding this continuum, the accuracy of these modalities, and treatment related outcomes based on imaging, will allow the clinician to counsel patients on management. This review will highlight the differences in conventional and molecular imaging in prostate cancer and how PSMA PET–CT can be used for the management of prostate cancer patients in different clinical scenarios, while providing cautionary notes for overtreatment.

## 1. Introduction:

Prostate cancer is the most common cancer and a leading cause of cancer-related deaths in men. There are an estimated 248,530 men diagnosed with this disease and 34,130 men died from metastatic prostate cancer in the United States in 2021 [1]. The two treatment modalities for initial treatment of prostate cancer include radical prostatectomy (RP) and/or brachytherapy or external beam radiation therapy. Following curative intent therapy, 5–50% of patients will develop biochemically recurrent prostate cancer (BCR) depending on Gleason Grade Group and curative treatment type utilized, equating to approximately 30,000–50,000 men diagnosed with BCR annually [2,3]. There exist several definitions of BCR depending on the specific population. The American Urological Association defines BCR after radical prostatectomy as a prostate specific antigen (PSA) level of at least 0.2 ng/mL, followed by a subsequent confirmatory PSA level of at least 0.2 ng/mL [4]. After radiation therapy, the American Society for Therapeutic Radiation and Oncology defines 3 successive PSA rises above the nadir as consistent with BCR. A consensus committee concluded that any rise in PSA levels of 2 ng/mL or more above the nadir, regardless of the type of radiation therapy given, is consistent with BCR (Phoenix Definition) [5]. For the past several years, BCR is typically the time when imaging is employed to evaluate local versus distant recurrence and to guide treatment.

Historically, conventional imaging, has been comprised of magnetic resonance imaging (MRI), computed tomography (CT) and bone scintigraphy using [^99m^Tc]Tc-methylene diphosphonate (MDP). The fact that patients would have negative conventional imaging despite rising PSA highlighted the lack of sensitivity of these imaging strategies. Thus, there have been sustained efforts over the years to improve the sensitivity of prostate cancer imaging. The introduction of molecular imaging, which includes ^11^C-choline position emission tomography (PET)–CT and ^18^F-fluciclovine (Axumin) PET–CT, has improved the detection of local and distant spread of prostate cancer (PC) [6]. ^11^C-choline PET–CT scans were FDA approved in 2012 for patients suspected of prostate cancer recurrence and non-informative conventional imaging [7]. ^18^F-fluciclovine PET–CT scans obtained FDA approval in 2016 for men with BCR after prior treatment [8]. The recent FDA approval of prostate-specific membrane antigen (PSMA) targeted agents [^68^Ga]Ga-PSMA-11 [9] and [^18^F]DCFPyl (Pylarify) [10] for evaluation of PC prior to definitive therapy as well as for BCR, afford a more powerful method to identify spread of PC and tailor management. Despite the approvals of these agents, there remains no clear clinical context about how they can be used to optimize care for recurrent disease.

The recent approval of [^18^F]DCFPyl (Pylarify) [10] and [^68^Ga]Ga-PSMA-11 [9] has been long awaited and brings hope that better detection can improve prostate cancer care. Compared to conventional imaging, PSMA PET radiopharmaceuticals have been shown to be more accurate, sensitive, and specific at identifying pelvic nodal or distant metastatic disease when used in primary staging of high-risk prostate cancer and lead to more management changes as seen in the ProPSMA study [11]. This new era of PC molecular imaging with PSMA PET–CT, introduces uncharted paths for the clinician to navigate. For example, with the diagnosis of metastatic disease occurring earlier with molecular imaging, how should the clinician interpret these findings since clinical trials to date have defined metastatic disease based on conventional imaging (CT/MRI and bone scintigraphy scan)? Does detecting metastatic disease via PSMA PET–CT lead to clinically meaningful outcomes with earlier initiation of treatment? As PSMA PET–CT imaging becomes widely available, it introduces new questions and the emergence of new terminology such as BCR with negative conventional imaging, BCR with positive conventional imaging, BCR with negative molecular imaging and BCR with positive molecular imaging. This terminology reveals a continuum of metastatic prostate cancer that the clinician can use to make clinically meaningful decisions for their patients. This review will highlight the differences in conventional and molecular imaging in prostate cancer and how PSMA PET–CT can be used for the management of prostate cancer patients in different clinical scenarios.

## 2. Conventional Imaging in Prostate Cancer

Conventional imaging for the diagnosis, surveillance and treatment assessment of prostate cancer includes MRI, CT and [^99m^Tc]Tc-MTD bone scintigraphy. These imaging modalities have been standard in clinical trials for decades in determining recurrence or progression and have been incorporated into recommendations by Prostate Cancer Clinical Trial Work Groups (PCWG). PCWG has evolved from PCWG1 (1999) which standardized PSA outcomes in castration-resistant prostate cancer to PCWG2 (2008), emphasizing drug efficacy assessment by control/relieve/eliminate or prevent/delay end points, such as spread to additional sites using conventional imaging [12]. Most recently PCWG3 (2016) provided further updates regarding clinical trial objectives in castrate resistant prostate cancer, however, continues to advise conventional imaging for baseline and progression assessment [12,13]. According to PCWG3, progression of nodal disease is considered when a previously normal lymph node grows by ≥5 mm in the short axis and is ≥1.0 cm, but must be ≥1.5 cm to be measurable, otherwise lymph node and visceral progression is based on Response to Treatment in Solid Tumors (RECIST 1.1). For bone disease, two new lesions represent progression, but only positivity on bone scans represents metastatic disease to the bones [13]. Location of disease, using conventional imaging, has been shown to impact prognosis of patients. In a meta-analysis investigating the impact of survival by metastatic castrate resistant prostate cancer disease site, the authors concluded lymph node-only disease had the best survival with a median overall survival of 31.6 months [14]. The other disease sites including non-visceral bone, lung and liver metastases had a median overall survival of 21.3, 19.4, and 13.5 months, respectively [14]. While conventional imaging has been instrumental in the management of prostate cancer, new imaging techniques have been investigated to improve diagnostic accuracy.

## 3. The New Era of PSMA PET Imaging

The introduction of molecular imaging into the field of prostate cancer has led to increased enthusiasm due to the ability to visualize smaller foci of prostate cancer earlier, compared to traditional imaging [15]. [^68^Ga]Ga-PSMA-11 (FDA approved December 2020) and [^18^F]DCFPyl (FDA approved May 2021) PET–CT scans were FDA approved for patients with suspected prostate cancer metastasis who are potentially curable by surgery or other therapy and for patients with suspected prostate cancer recurrence based on elevated PSA levels [16]. PSMA is a type II integral membrane glycoprotein that is highly expressed in prostate secretory-acinar epithelium and prostate cancer, as well as in several extraprostatic tissues [17]. Multiple studies have shown that PSMA is expressed in tumor-associated neovasculature of many tumor types besides prostate cancer, including glioblastoma, breast, colorectal, and renal [18,19]. PSMA is highly expressed in the vascular endothelium of various malignancies but is not evident in normal vascular endothelium [20]. High PSMA expression is associated with high-grade prostate cancer groups and increases in lymph node metastases, recurrent disease and distant metastases compared to primary tumors [17]. While elevated PSMA expression of the primary tumor at the time of prostatectomy is associated with higher Gleason score and PSA at diagnosis, PSMA is not an independent predictor of lethal prostate cancer [21]. Therefore, its role in predicting outcomes is unclear; however, PSMA’s high expression in prostate cancer has proven it to be a successful target for prostate cancer specific imaging and targeted radionuclide therapy.

In the proPSMA trial [^68^Ga]Ga-PSMA-11 PET–CT was compared to conventional imaging in patients with high-risk prostate cancer before curative intent surgery or radiotherapy. In this randomized study, the primary outcome was identifying either pelvic nodal or distant metastatic disease and [^68^Ga]Ga-PSMA-11 PET–CT outperformed conventional imaging in accuracy (92% vs. 65%; *p* < 0.0001), sensitivity (85% vs. 38%) and specificity (98% vs. 91%), respectively [11]. The diagnostic performance of [^18^F]DCFPyl PET–CT was evaluated in the OSPREY trial in two cohorts: cohort A (initial staging high risk prostate cancer) and cohort B (detection of recurrent prostate cancer). In cohort A, when lymph nodes were greater than 5 mm, the sensitivity was 60%, specificity 97.9%, positive predictive value 84.6%, and negative predictive value 92.2%. In cohort B with BCR, the PPV was 95.8% and NPV was 81.9% [22]. Historically, CT imaging diagnostic performance in prostate cancer has a specificity of 82%, PPV of 32%, and NPV of 12%. Therefore, [^18^F]DCFPyl PET–CT appears to outperform conventional imaging, except in sensitivity where results are similar [22]. With improved diagnostic performance compared to conventional imaging, these two PSMA-PET–CT scans can detect advanced prostate cancer earlier, which can lead to management change. Further studies are needed to evaluate and validate the utility of PSMA PET–CT in this setting.

### 3.1. Redefining Prostate Cancer Recurrence with PSMA PET Imaging

The presence of metastatic prostate cancer, which was once determined based on conventional imaging, is now being challenged with the introduction of PSMA PET scans and earlier detection of metastatic disease that had previously been considered subclinical. In a sub-analysis of the OSPREY trial ([^18^F]DCFPyl PET–CT), 58% (*n* = 19) of patients were upstaged from M0 by conventional imaging to M1 by PSMA PET–CT (10 extra-pelvic lymph node and 9 bone lesions) [23]. As non-regional lymph nodes, bone or visceral metastases are revealed without conventional imaging correlates, it suggests metastatic disease based on imaging techniques exists on a continuum on which a patient transitions from BCR to metastatic disease (Figure 1). Risk stratifying BCR patients with terminology such as: (1) BCR with negative conventional imaging and negative PSMA PET–CT imaging, (2) BCR with positive PSMA PET–CT and negative conventional imaging, or (3) BCR with positive PSMA PET–CT and positive conventional imaging, will aid the clinician in counseling the patient and developing a therapeutic strategy. It remains unclear if non-specific abnormalities on CT will be overcalled given findings on PSMA PET imaging.

In the CONDOR trial, [^18^F]DCFPyl PET–CT was performed in patients with biochemical recurrence post RP with a median PSA of 0.8 ng/mL and 63.9% of patients had a change in intended management after PSMA PET imaging [24]. Some of the notable management changes out of 131 patients included salvage local therapy to systemic therapy (*n* = 58, 44.3%), non-curative systemic therapy to salvage (*n* = 43, 32.8%), observation to initiating therapy (*n* = 49, 37.4%) and planned treatment to observation (*n* = 9, 6.9%). These management changes have clinical importance as some patients were able to undergo curative intent salvage radiation treatment, when previously they would have been treated systemically. Other patients avoided systemic therapy’s adverse effects as their treatment was changed to observation. However, the patients that were changed from observation to systemic treatment may have been overtreated with unclear survival benefit since outcomes based on PSMA PET–CT are not clearly defined. Clinicians will need to counsel patients regarding how this imaging modality will impact management decisions, which includes survival outcome uncertainty and early initiation of systemic treatment with the potential to impact quality of life. 

### 3.2. Interpreting PSMA PET Scans to Guide Management

The efficacy of diagnostic imaging can be determined based on diagnostic accuracy (sensitivity, specificity), effect on treatment (treatment planning, changes in management), patient health outcomes and cost effectiveness [25]. Determining patient health outcomes is generally delayed due to cost and the number of patients needed to obtain meaningful results [25]. PSMA PET–CT is in its early stages of efficacy hierarchy with completion of diagnostic accuracy and some studies showing effects on treatment. The true value of this diagnostic imaging modality will be determined based on improvement of patient outcomes (survival, toxicity) when used prior to definitive prostate cancer treatment, or in the biochemical recurrent setting. For example, a positive PSMA PET–CT prior to definitive surgery would lead to alternative treatment strategies sparing the patient from a surgery that would offer no benefit and potentially causing harm, while the impact of a positive PSMA PET–CT prior to definitive radiation (potentially with ADT) or positive in the setting of BCR remains unclear.

As PSMA PET–CT becomes more widely available, clinicians will be required to interpret the results to guide management and provide appropriate patient counseling. If conventional imaging and PSMA PET–CT imaging findings agree, then the clinician can use the standard of care based on NCCN guidelines. However, if conventional imaging and molecular imaging with PSMA PET–CT are conflicting then interpreting these results requires an understanding of evidence to date to assist in decision-making.

#### 3.2.1. Scenario 1: A Patient Presents with Biochemical Recurrent Prostate Cancer with Positive Conventional Imaging and Negative PSMA PET–CT Imaging

Patients with conventional imaging suggestive of regional lymph node disease, non-regional lymph node disease, or distant metastatic disease may undergo further PSMA PET–CT imaging to guide treatment. CT imaging, when performed for lymph node staging, has a positive predictive value of 32% [26]. For prostate cancer bone metastases, bone scintigraphy has a PPV of 45% [27], both suggesting that false positive results frequently occur. If a PSMA PET–CT is negative for lymph nodes or distant disease while conventional imaging is suggestive of spread, the clinician will need to understand the sensitivity and NPV of PSMA scans to guide management and counseling of the patient. The sensitivity of [^18^F]DCFPyl PET–CT for pelvic lymph nodes, extra-pelvic lymph nodes, bone and visceral/soft tissue disease is 100% (*n* = 15), 96.4% (*n* = 56), 96.8% (*n* = 43), and 100% (*n* = 10), respectively, and the NPV is 81.9% [22], therefore, based on these studies, some patients will be missed for having more advanced disease with a negative PSMA PET–CT scan. However, other studies have investigated discordance between PSMA imaging and other molecular imaging types including 2-[^18^F]FDG PET imaging. 2-[^18^F]FDG PET imaging for prostate cancer staging and the detection of recurrence is not recommended by NCCN guidelines based on the limited studies available and inconsistent results. In a study of 41 men with a Gleason score ≥ 8 who underwent 2-[^18^F]FDG PET for staging and were found to have nodal metastases on histopathology (*n* = 11), only 27% (*n* = 3) had corresponding FDG lymph node uptake which highlights the limitations with 2-[^18^F]FDG PET in prostate cancer [28]. A systematic review was performed by McGeorge et al., comparing PSMA and 2-[^18^F]FDG PET for staging prostate cancer. In this review, when 2-[^18^F]FDG PET was performed after PSMA PET, the detection of metastases improved from 65% to 73% in high-risk early castration resistant prostate cancer with negative conventional imaging. A positive 2-[^18^F]FDG PET was found in 17% of men with a negative PSMA for postprostatectomy biochemical recurrence, highlighting a potential benefit of combining molecular tracers [29]. In the TheraP trial, investigating 177Lu-PSMA-617 versus cabazitaxel in patients with metastatic castration resistant prostate cancer (mCRPC), 18% of patients had discordant FDG-positive and PSMA-negative findings [30]. While this study was in mCRPC, it suggests that metastatic prostate cancer could be FDG positive while being PSMA negative, lending to the possibility of missing metastatic disease lesions. Discordance may also occur between choline and PSMA PET–CT. In a study of 67 patients with BCR, 6% of lymph nodes were choline only positive and 2% of patients had choline only lymph node disease. Therefore, very few patients would be upstaged with the addition of choline PET imaging in the setting of negative PSMA PET. When comparing Na[^18^F]F PET–CT and PSMA PET–CT, the investigators appreciated considerable discordance and revealed PSMA activity in prostate cancer metastases and bone turnover becomes weaker in more advanced stages of disease [31]. This suggests Na[^18^F]F PET–CT could offer further guidance in determining the presence of metastatic bone lesions in the setting of a negative PSMA PET–CT. More studies are needed in combining imaging modalities to improve the false negative rate; however, this may lead to higher expense. Based on current studies, patients with BCR prostate cancer with positive conventional imaging and negative PSMA PET–CT imaging can undergo biopsy or BCR management options which include clinical trial, observation, SRT, or intermittent ADT (Figure 2), however, repeat molecular imaging with a different molecular tracer could be considered in the appropriate clinical context.

#### 3.2.2. Scenario 2: A Patient Presents with Biochemical Recurrence with Negative Conventional Imaging and Positive Local or Regional Lymph Nodes (N1M0) by PSMA PET–CT Imaging

For prostate cancer recurrence, the positive predictive value of [^18^F]DCFPyl PET–CT for pelvic lymph nodes is 77.8% (*n* = 18), indicating that positive results are consistent with prostate cancer recurrence [22]. In the recurrent prostate cancer setting, smaller studies using PSMA PET–CT have been published showing its clinical benefits that may translate to improved outcomes. [^68^Ga]Ga-PSMA-11 PET–CT has been shown to independently predict treatment response of BCR patients to salvage radiation therapy (SRT) with response defined as both PSA ≤ 0.1 ng/mL and >50% reduction in PSA [32]. With a limited median follow-up time of 10.5 months, 99 patients who developed biochemical recurrence after radical prostatectomy received SRT. A total of 86% (*n* = 23/27) of patients with a negative PSMA PET–CT achieved a response to SRT with only 10% (*n* = 3/29) having PSA failure. This compared to those patients with PSMA negative disease that did not undergo SRT and had a PSA rise 65% (22/34) of the time. This suggests that patients with negative PSMA PET–CT could benefit from SRT. The response to SRT was approximately 25% less if nodal disease was seen on PSMA scans. If nodal disease was positive via PSMA PET–CT, 61.5% of patients (*n* = 16/26) had a treatment response to SRT and 38.5% (*n* = 10/26) developed biochemical progression despite SRT in the 10.5 month median follow up time. As expected, response to SRT was even less if distant disease was seen via PSMA PET–CT, with 30% (*n* = 3/10) achieving a response to RT and the majority (70%) developing biochemical progression despite RT [32]. While followup is limited and the number of patients is low, this offers insight into how this imaging modality could be used to guide treatment and further counsel patients. However, the survival benefits from using PSMA PET–CT are lacking prospective evidence and further studies will be necessary to see if outcomes are similar to SRT management of pelvic lymph node recurrence when detected by conventional imaging, which have a 5-year overall survival of 82.5%. In relation to local recurrence, it is important to note that CT imaging may miss local recurrence, or when performed with PSMA tracers, can overcall local recurrence. This was seen in a prospective study by Tulsyan et al., who showed multiparametric MRI may better detect local relapse than PSMA PET–CT since concordance between the two imaging modalities was 52% [33]. For example, when [^68^Ga]Ga-PSMA-11 PET–CT was used it over-reported seminal vesical involvement, bladder and rectum invasion and radiologists could not comment on capsular involvement. Therefore, clinicians should take this into consideration when evaluating local recurrence. In patients with negative conventional imaging and positive PSMA PET–CT in prostate fossa alone, or regional LNs then the clinician should counsel the patient on undergoing SRT.

#### 3.2.3. Scenario 3: Patient Presents with Biochemical Recurrence with Negative Conventional Imaging and Positive Extra-Pelvic Lymph Nodes (M1a), Bone (M1b), or Visceral Metastases (M1c) by PSMA PET–CT Imaging

This clinical scenario will be frequent since a sub-analysis of the OPSREY trial ([^18^F]DCFPyl PET–CT) showed 58% (*n* = 19) of patients were upstaged from M0 by conventional imaging to M1 by PSMA PET–CT (10 extra-pelvic lymph node and 9 bone lesions) [23]. The positive predictive value of [^18^F]DCFPyl PET–CT for extra-pelvic lymph nodes, bone, and visceral disease is 83.1% (*n* = 65), 81.6% (*n* = 44) and 90% (*n* = 10), respectively [22]. In the setting of limited metastases, an oligometastatic treatment approach may be considered.

The great hope is that PSMA imaging will be able to significantly impact the course of disease in these patients. In the randomized STOMP trial, surveillance, or metastasis-directed therapy for oligometastatic prostate cancer (three lesions or less) seen on choline PET–CT, was evaluated. Although the five-year ADT-FS was statistically significant favoring MDT when compared to surveillance, 34% vs. 8%, respectively, this highlights the fact that even in patients with limited disease, PET directed therapy is not likely to be curative as it likely still does not show all the metastatic disease [34]. In another study of 114 patients with biochemical failure, PSMA positive lesions were treated with radiosurgery and the patients with low PSA and a single PSMA lesion were found to benefit the most, while patients with multiple PSMA positive lesions had a biochemical recurrence within a median of eight to twelve months [35]. These data do however, highlight the potential benefit of PET directed therapy for patients with limited disease on PSMA scans on a case-by-case basis.

If PSMA PET–CT imaging is positive without conventional correlates that would have been independently seen on CT or [^99m^Tc]Tc-MTD bone scintigraphy, there may be a knee jerk reaction to label them as metastatic in the conventional sense. Then clinicians will be considering agents approved in the metastatic castration sensitive prostate cancer (mCSPC) such as docetaxel, apalutamide, enzalutamide, or abiraterone. The outcomes data for this patient population is unclear since PSMA PET positive “metastatic” patients would not have met criteria for trial enrollment for these agents in mCSPC [36,37,38,39]. All these trials in mCSPC would have excluded patients with only PET positive disease and negative CT and [^99m^Tc]Tc-MTD bone scintigraphy. Furthermore, the evaluation of abiraterone (LATTITUDE) excluded all patients with recurrent disease while the studies with docetaxel (CHAARTED) and enzalutamide (ENZAMET) all had confidence intervals that crossed 1.0. Thus, upon closer review of these studies their application to PSMA PET positive disease is likely to introduce lead time bias, without clear benefits in disease progression or survival. Meaning, earlier detection with initiation of treatment may lead to longer perceived survival time, when in actuality, the survival time is unchanged if more advanced disease was found later with later initiation of treatment.

Understanding the natural history of BCR based on existing data should be used to inform clinicians about their management decisions in patients who have conventional imaging devoid of metastatic disease despite PSMA scan findings. Before the introduction of molecular imaging, clinicians would often manage BCR with surveillance, salvage radiotherapy or ADT while monitoring for metastatic disease to present on conventional imaging. In a retrospective review by Antonarakis et al., 450 patients treated with prostatectomy and developed BCR and did not undergo adjuvant or salvage radiotherapy were assessed for metastasis free survival (MFS). The median metastasis free survival was ten years and 29.8% of patients (134 of 450) developed metastatic disease [40]. In a multivariate analysis, PSA doubling time and Gleason score were independent predictors of MFS. The median MFS for Gleason score 8–10 and Gleason score 7 was four years and eleven years, respectively. Based on a PSA doubling time of <3 months, 3–8.9 months, 9–14.9 months or ≥15 months, the median MFS was 1, 4, 13, 15 years, respectively. To this end, it is also worth noting that the STOMP trial had a confidence interval that crossed 1.0 in patients who had a PSA doubling time greater than three months [34].

Given that most patients in this Antonarakis et al., analysis of PSA doubling time and metastasis free survival, had a PSA > 0.5, it would be expected that these patients all lived many years with subclinical disease that could have been seen on PSMA scans had the technology been available. Yet, in the absence of such imaging, they likely lived years not just free of metastasis, but also free of toxicity related to ADT plus anti-androgen, or chemotherapy. When compared to the survival timelines seen in the mCSPC trials, we start to understand the potential extent of lead time bias. While patients with a PSA doubling time beyond three months may be expected to have a median four-year metastasis-free survival, patients from the mCSPC trials often had a four to five year overall survival [36,37,38,40]. The biology of recurrent prostate cancer is likely much more indolent than mCSPC and equating these populations (and thus their clinical management) is not actually supported by current data. Furthermore, previous studies of ADT alone in these recurrent populations did not demonstrate an advantage for ADT alone compared to ADT deferred until metastasis [41]. And while the advent of ADT based combinations may be more effective, prospective data is required given the cost and potential prolonged extra exposure to ADT combinations that could result from incorrectly equating PSMA scan positive BCR with mCSPC. In addition, future prospective studies or retrospective reviews are needed to determine if androgen signaling inhibitors used earlier translate to improved outcomes, as expected. If polymetastatic disease is revealed with PSMA PET–CT, then the clinician should risk stratify the patient based on disease burden, PSA level, PSA doubling time, and comorbidities to determine if the benefits of initiating systemic therapy with agents approved for mCSPC outweighs the risks of toxicity and lead time bias.

**Figure 2 cancers-14-01361-f002:**
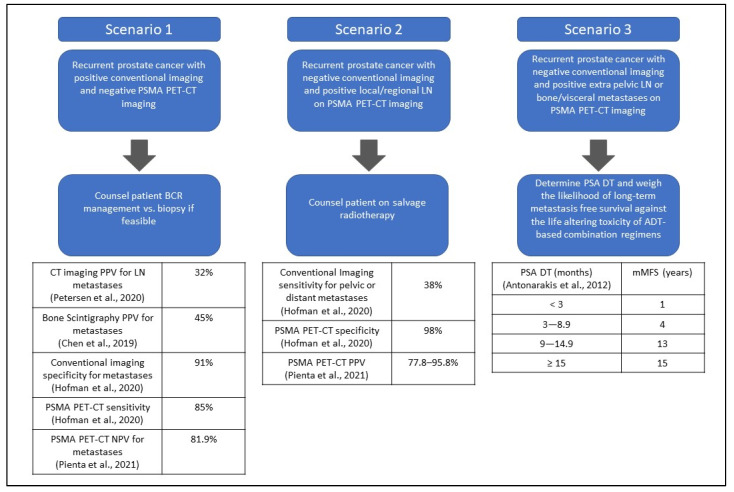
Summary of Clinical Scenarios [11,22,26,27,40]. The median metastasis free survival (mMFS) based on PSA DT for BCR prostate cancer and negative conventional imaging.

## 4. Conclusions

PSMA PET–CT imaging will become incorporated into everyday practice for prostate cancer management and requires the clinician, in collaboration with a nuclear medicine physician, to interpret the findings appropriately when recent therapeutic advances have been approved in a conventional imaging era. While the benefits of more sensitive imaging in newly diagnosed disease is clear when deciding if curative therapy (e.g., surgery or radiation) is feasible and appropriate, the impact on patients with recurrent disease after definitive therapy may be more ambiguous. PSMA-based PET imaging reveals a continuum of metastatic disease that can result in earlier initiation of therapy with potential survival benefits, or lead to overtreatment impacting quality of life. If conventional imaging and PSMA PET–CT imaging findings agree, then standard of care is provided. However, if conventional imaging and PSMA PET–CT imaging results are conflicting, interpreting these results with the knowledge of retrospective studies will be required until prospective evidence is available to prevent overtreatment and optimize patient care. Upon reviewing existing data, it becomes more evident that caution should be used when labeling PSMA positive patients as equivalent to mCSPC and equating them to populations in mCSPC trials, when it is very likely that the indolent nature of recurrent disease may not be as impacted from treatment intensification of ADT-based combinations. Deploying such ADT combinations earlier may have more ambiguous net benefits to patients when weighed against the toxicity of such regimens which may need to be administered for a decade or more in some cases.

## Figures and Tables

**Figure 1 cancers-14-01361-f001:**
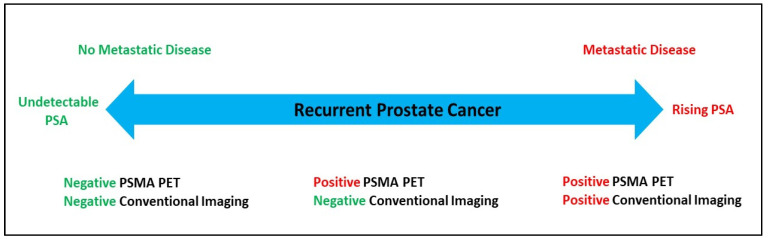
Continuum of metastatic disease in the setting of PSMA PET–CT.

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
