# Peer review of "The Continuum of Metastatic Prostate Cancer: Interpreting PSMA PET Findings in Recurrent Prostate Cancer"

_cancers, 2022, doi:10.3390/cancers14061361_

Round 1

Reviewer 1 Report

The authors describe the potential benefits and disadvantages of the use of PSMA-PET/CT in the recurrent setting of prostate cancer. They give a good overview of the existing literature about diagnostic test characteristics of conventional vs PSMA imaging. And they describe some scenario's in which there is discrepancy between these two modalities. These scenario's give a good insight in the way PSMA imaging might influence change of management. And in the problems that we face because of lack of randomized trials in this area. 

Author Response

Thank you Review 1 for your comments. We have reviewed the manuscript and corrected the minor style and spelling errors as requested. 

Reviewer 2 Report

cancers-1604782

The present narrative review by Kase et al. highlights the differences in conventional and molecular imaging in prostate cancer and how PSMA PET-CT can be used to manage prostate cancer patients in different clinical scenarios.   

The topic of the review is undoubted of high clinical interest. While the added diagnostic accuracy of PSMA PET/CT compared to conventional imaging technologies has been extensively investigated, the impact of next-generation imaging on clinical management and the consequent improvement in the clinical outcomes still represent an unmet need.  

The paper is well structured, clearly written, and the available literature is rigorously and critically analyzed.

Just a few minor considerations from my side:

  • The names of the radiopharmaceuticals should be used according to the EANM Guidance to Radiotracer Nomenclature https://www.eanm.org/publications/guidelines/nomenclature/.
  • Introduction, line 75: “The recent approval of F-18 DCFPyL (Pylarify) [10] has been long-awaited and brings hope that better detection can improve prostate cancer care”. This sentence can be extended to both DCFPyL and PSMA-11.
  • Introduction, lines 76-79: please specify that these findings were obtained in the setting of the primary staging of hrPCa patients.
  • Chapter 2, line 114: please add ref [14] at the end of the sentence.
  • Chapter 3, line 161, please correct “OSPREY”.
  • Chapter 4, line 383: “nuclear medicine physician” is generally preferred compared to “nuclear medicine radiology specialist”.

Author Response

Thank you to the reviewer for their expert responses and suggestions. All of the comments have been addressed. 

  • The names of the radiopharmaceuticals should be used according to the EANM Guidance to Radiotracer Nomenclature https://www.eanm.org/publications/guidelines/nomenclature/.--Completed
  • Introduction, line 75: “The recent approval of F-18 DCFPyL (Pylarify) [10] has been long-awaited and brings hope that better detection can improve prostate cancer care”. This sentence can be extended to both DCFPyL and PSMA-11.—Completed
  • Introduction, lines 76-79: please specify that these findings were obtained in the setting of the primary staging of hrPCa patients. –Completed
  • Chapter 2, line 114: please add ref [14] at the end of the sentence. --Completed
  • Chapter 3, line 161, please correct “OSPREY”. –Completed
  • Chapter 4, line 383: “nuclear medicine physician” is generally preferred compared to “nuclear medicine radiology specialist”.—Completed